# Multi-view clustering via global-view graph learning

**Qin Li**[ID], **Geng Yang***

School of Computer and Software Engineering, Shenzhen Institute of Information Technology, Shenzhen, China

* yangg@sziit.edu.cn

## Abstract

Multiview clustering aims to improve clustering performance by exploring multiple representations of data and has become an important research direction. Meanwhile, graph-based methods have been extensively studied and have shown promising performance in multiview clustering tasks. However, most existing graph-based multiview clustering methods rely on assigning appropriate weights to each view based on its importance, with the clustering results depending on these weight assignments. In this paper, we propose an a novel multiview spectral clustering framework with reduced computational complexity that captures complementary information across views by optimizing a global-view graph using adaptive weight learning. Additionally, in our method, once the Global-view Graph is obtained, cluster labels can be directly assigned to each data point without the need for any post-processing, such as the K-means required in standard spectral clustering. Our method not only improves clustering performance but also reduces computational resource consumption. Experimental results on real-world datasets demonstrate the effectiveness of our approach.

**Data availability statement:** All data files are available from the MSRC-v1, Caltech-101, ORL, and Yale databases via https://mldta.com/dataset/msrc-v1/, https://tensorflow.google.cn/datasets/catalog/caltech101, http://www.uk.research.att.com/facedatabase.html, and

## Introduction

In many practical applications such as video surveillance and image retrieval, heterogeneous features representing the same instance can be obtained. For example, images can be represented by various descriptors such as SIFT [1], HOG [2], GIST [3], and LBP [4]; webpages can be described by their content, the text of the pages that link to them, and the link structure of the linked pages. Since these heterogeneous features summarize the characteristics of objects from different perspectives, they are considered as multiple views of the data. Multiview learning, which aims to explore information from different views to improve learning performance, has become an important research direction [5,6].

Clustering is an unsupervised learning task that divides objects into meaningful groups. To appropriately integrate information from multiple views in clustering, many multiview clustering methods have been proposed [7–10]. These methods can be broadly categorized into three main approaches: spectral clustering-based methods, subspace clustering-based methods, and other advanced techniques such as tensor-based clustering.

http://vision.ucsd.edu/content/yale-face-database.

**Funding:** This work was supported by Natural Science Foundation of Guangdong Province under Grant 2023A1515011845.

**Competing interests:** The authors have declared that no competing interests exist.

### Spectral clustering-based methods

Spectral clustering has become one of the most popular modern clustering algorithms. It is easy to implement, can be efficiently solved using standard linear algebra software, and generally outperforms traditional clustering algorithms such as k-means. Graph-based multi-view clustering algorithms, which extend spectral clustering to handle multi-view data, have shown good performance and have been widely studied. Kumar et al. [11] extended spectral clustering by co-regularizing the clustering assumptions of different views, ensuring that the graphs from different views are consistent with each other. Cai et al. [12] developed a multimodal spectral clustering algorithm to learn a common Laplacian matrix by imposing a non-negative constraint on the relaxed clustering assignment matrix. Nie et al. [13] proposed the Auto-weighted Multiple Graph Learning (AMGL) algorithm, which automatically learns a set of weights for all graphs without additional parameters. Xia et al. [14] proposed a Markov chain-based method for robust multi-view spectral clustering (RMSC). Based on bipartite graphs, Li et al. [15] used local manifold fusion to integrate heterogeneous features and proposed a new large-scale multi-view spectral clustering method (MVSC).

### Subspace clustering-based methods

Subspace clustering-based approaches focus on finding a low-dimensional subspace that captures the shared structure across multiple views. Zhan et al. [16] introduced Multiview Consensus Graph Clustering (MCGC), which combines consensus matrix learning with subspace clustering to enhance clustering performance. Luo et al. [17] proposed Consistent and Specific Multi-view Subspace Clustering (CSMSC), which models both the consistency and specificity of multi-view data to learn a shared subspace representation. While these methods are effective in leveraging complementary information across views, they often face challenges with computational complexity and scalability.

### Constrained Laplacian Rank (CLR) methods

The Constrained Laplacian Rank (CLR) model [18] is a significant advancement in graph-based clustering. CLR constructs a similarity matrix by imposing a rank constraint on the Laplacian matrix, ensuring that the resulting similarity matrix has exactly $c$ connected components (where $c$ is the number of clusters). This guarantees that the clusters are clearly separated in the graph structure.

Unlike traditional spectral clustering methods that require additional post-processing such as k-means, CLR directly assigns cluster labels to each data point. This avoids potential errors introduced by post-processing and improves efficiency. However, CLR was initially designed for single-view clustering and cannot simultaneously handle information from multiple views.

To address this, Nie et al. [19] extended CLR to multi-view clustering by proposing methods such as Parameter-Weighted Multi-view Clustering (PwMC) and Self-Weighted Multi-view Clustering (SwMC). While effective, these methods have certain limitations, such as sensitivity to weight parameters and the inability to fully integrate global and complementary information from multiple views.

### Tensor analysis

Recent developments in tensor analysis and deep learning have inspired advanced methods for multi-view clustering. Tensor analysis-based methods [20–22] exploit the higher-order structure of multi-view data to capture relationships between views, achieving promising results. However, these methods generally involve high computational complexity.

Although these approaches demonstrate effectiveness in specific scenarios, they still have certain limitations. Most methods fail to fully integrate the complementary information between views and the global structure of the data.

## Our contributions

To address these limitations, we propose a new method based on CLR: Multi-view Clustering via Global-view Graph Learning (MCGGL). The proposed method constructs a Global Affinity Matrix that simultaneously captures the specificity of each view and the complementary information between different views. Unlike traditional methods, our approach directly optimizes the global similarity graph, enabling clustering label assignment without the need for post-processing. This results in improved clustering performance and computational efficiency. Experimental results on real-world datasets demonstrate the effectiveness of our proposed method.

## Notations

In the entire text, all matrices are denoted by uppercase letters. For a matrix $M$, the $i$th row and the $i,j$th element of $M$ are denoted by $m_i$ and $m_{ij}$, respectively. The trace of $M$ is denoted as $\text{Tr}(M)$. The $v$th view of matrix $M$ is denoted as $M^{(v)}$. The transpose of matrix $M$ is denoted as $M^T$. The L2-norm of a vector $v$ is denoted as $\|v\|_2$, and the Frobenius norm of a matrix $M$ is denoted as $\|M\|_F$. Specifically, we use $1_n$ to represent an $n$-dimensional column vector where each element is 1.

## Related work

### CLR clustering

Given an initial affinity matrix $A \in \mathbb{R}^{n \times n}$, CLR aims to learn a new similarity matrix $S \in \mathbb{R}^{n \times n}$ that has exactly $c$ connected components, where $n$ is the number of data points and $c$ is the number of clusters. The Laplacian matrix associated with $S$ is defined as $L_S = D_S - \frac{S^T + S}{2}$, where $D_S$ is a diagonal matrix with its $i$th diagonal element being $\sum_j \frac{s_{ij} + s_{ji}}{2}$.

The Laplacian matrix has an important property as follows [23]:

**Theorem 1**: The multiplicity of the eigenvalue 0 in the Laplacian matrix $L_S$ is equal to the number of connected components in the graph associated with $S$.

Based on this observation, Nie et al. [18] constrained the rank of $L_S$ to be $n - c$ and proposed the following CLR model for graph clustering based on L1 norm and L2 norm distances:

$$J_{CLR-L1} = \min_{\sum_j s_{ij}=1, s_{ij} \geq 0, rank(L_S)=n-c} \|S - A\|_1 \tag{1}$$

$$J_{CLR-L2} = \min_{\sum_j s_{ij}=1, s_{ij} \geq 0, rank(L_S)=n-c} \|S - A\|_F^2 \tag{2}$$

However, these problems seem difficult to solve because $L_S = D_S - \frac{S^T + S}{2}$, and $D_S$ also depends on $S$, while the constraint $rank(L_S) = n - c$ is a complex nonlinear constraint. Nie et al. [18] proposed a novel and effective algorithm to address these issues.

Let $\delta_i(L_S)$ denote the $i$th smallest eigenvalue of $L_S$. Note that $\delta_i(L_S) \geq 0$ since $L_S$ is positive semidefinite. Problem (1) is equivalent to the following problem for sufficiently large values of $\lambda$:

$$\min_{\sum_j s_{ij}=1, s_{ij} \geq 0} \|S - A\|_F^2 + 2\lambda \sum_{i=1}^{c} \delta_i(L_S) \tag{3}$$

When $\lambda$ is sufficiently large, $\delta_i(L_S) \geq 0$ holds for each $i$, so the optimal solution $S$ to problem (3) will make the second term $\sum_{i=1}^{c} \delta_i(L_S)$ equal to zero, thereby satisfying the constraint $\text{rank}(L_S) = n - c$ in problem (1).

According to Ky Fan's theorem [24],

**Theorem 2**: For a Hermitian matrix, the sum of the smallest $k$ eigenvalues is equal to the minimum trace of the matrix formed by projecting onto any $k$-dimensional subspace.

This give us:

$$\sum_{i=1}^{c} \delta_i(L_S) = \min_{H \in \mathbb{R}^{n \times c}, H^T H = I} Tr(H^T L_S H) \tag{4}$$

Thus, problem (3) is further equivalent to the following problem:

$$\min_{H^T H = I} \|S - A\|_F^2 + 2\lambda \text{Tr}(H^T L_S H)$$
$$\text{s.t.} \sum_{j} s_{ij} = 1, \ s_{ij} \geq 0, \ H \in \mathbb{R}^{n \times c}, \ H^T H = I \tag{5}$$

Compared to the original problem (1), problem (5) is easier to solve. It has been demonstrated that CLR achieves superior performance in clustering [18]. Note that the CLR method is applicable only to single-view data; it cannot simultaneously handle graphs from multiple views.

## Parameter-weighted multi-view clustering (PwMC)

CLR is a single-view graph-based clustering method. Nie et al. [19] extended this technique to the field of multi-view clustering. For multi-view data, let $m$ be the number of views, and $A^{(1)}, A^{(2)}, \ldots, A^{(m)}$ be the corresponding input sample matrices, where $A^{(v)} \in \mathbb{R}^{n \times n}$ ($1 \leq v \leq m$). Each view should be assigned a weight to measure its importance, and this idea can be naturally modeled by minimizing a linear combination of the reconstruction error $\|S - A^{(v)}\|_F^2$ for each view. Therefore, the constructed objective can be written as:

$$\min_{S, \alpha^v} \sum_{v=1}^{m} \alpha^v \|S - A^{(v)}\|_F^2 + \gamma \|\alpha\|_2^2$$
$$\text{s.t.} \alpha^v \geq 0, \alpha^T 1_m = 1, s_{ij} \geq 0, \sum_{j} s_{ij} = 1, \text{rank}(L_S) = n - c. \tag{6}$$

where $\alpha = [\alpha^1, \alpha^2, \ldots, \alpha^m]^T$ and $\gamma > 0$. The second term in problem (3) is used to smooth the weight distribution. The constraints $s_{ij} \geq 0$ and $\sum_{j} s_{ij} = 1$ ensure that $S$ is a non-negative matrix and each row represents a normalized probability distribution, maintaining the validity and stability of the similarity matrix.

PwMC involves an undesirable parameter $\gamma$. In an unsupervised learning setting without labeled instances, $\gamma$ cannot be obtained through traditional supervised hyperparameter tuning techniques such as cross-validation. Additionally, it was observed that the final experimental performance is very sensitive to $\gamma$, and the optimal value of $\gamma$ varies across different datasets, making PwMC (and related multi-view clustering methods that use similar weight learning strategies) impractical. Therefore, to remove $\gamma$ while retaining much of the accuracy, Nie et al. [19] further proposed a new self-weighted multi-view clustering (SwMC) method.

## Self-weighted multi-view clustering (SwMC)

For Self-Weighted Multi-View Clustering (SwMC), the proposed objective function is:

$$\min_{s_{ij} \geq 0, \sum_j s_{ij}=1, \text{rank}(L_S)=n-c} \sum_{v=1}^{m} \|S - A^{(v)}\|_F \tag{7}$$

After a series of transformations, Nie et al. [19] rewrote equation (7) into the following form, allowing the weight parameters $w^{(v)}$ to be learned adaptively:

$$\min_{s_{ij} \geq 0, \sum_j s_{ij}=1, \text{rank}(L_S)=n-c} \sum_{v=1}^{m} w^{(v)} \|S - A^{(v)}\|_F^2 \tag{8}$$

## Proposed method

### Problem formulation and objective function

To effectively capture the complementary information between different views while considering the specific information of each view to obtain optimal global information, we designed the following multi-view clustering objective function.

$$\min_{S} \quad \sum_{v=1}^{m} \|S - A^{(v)}\|_F + \beta \|S - A\|_F$$
$$\text{s.t.} \quad \sum_j s_{ij} = 1, \quad s_{ij} \geq 0, \quad \text{rank}(L_S) = n - c. \tag{9}$$

In our model, $i$ represents the $i$th row, $j$ represents the $j$th column, $m$ represents the number of views. Specifically, based on the initial affinity matrix $A^{(v)}$ of each given view, we constructed a Global Affinity Matrix $A$ and a Global Similarity Matrix $S$. We then capture the complementary information between different views by minimizing the distance between $S$ and $A$, and capture the specific information of each view by minimizing the distance between $S$ and $A^{(v)}$. By analyzing the Global-view Graph corresponding to $S$, clustering labels can be directly assigned to each data point without any post-processing, such as the K-means step required in standard spectral clustering.

$A^{(v)} \in \mathbb{R}^{n \times n}$ represents the initial affinity matrix learned from the sample matrix $X^{(v)}$ of the $v$th view. $A \in \mathbb{R}^{n \times n}$ represents the initial affinity matrix learned from the global sample matrix $X$. The method for calculating the initial affinity matrix follows the standard CLR model [18].

The global sample matrix is obtained by stacking the sample matrices $X^{(v)}$ of each view. For example, if there are three sample matrices $X^{(1)}$, $X^{(2)}$, and $X^{(3)}$ from three views, we convert each sample matrix into a column vector, forming a global sample matrix $X = [X^{(1)} X^{(2)} X^{(3)}]^T \in \mathbb{R}^{(d_1+d_2+d_3) \times n}$.

$S$ is the global similarity matrix to be solved. The Laplacian matrix associated with $S$ is defined as $L_S = D_S - S$, where $D_S$ is a diagonal matrix with diagonal elements $\sum_j (s_{ij} + s_{ji})/2$. $H$ is the clustering label matrix to be solved.

Specifically, the complementary information is captured by minimizing the discrepancy between the Global Similarity Matrix $S$ and both the Global Affinity Matrix $A$ and view-specific affinity matrices $A^{(v)}$. Mathematically, the objective function can be expressed as:

$$\min_{S} \quad \sum_{v=1}^{m} W_v \|S - A^{(v)}\|_F^2 + \beta W \|S - A\|_F^2$$
$$\text{s.t.} \quad \sum_{j} s_{ij} = 1, \quad s_{ij} \geq 0, \quad \text{rank}(L_S) = n - c. \tag{10}$$

Here, $W_v$ and $W$ are adaptive weights that dynamically adjust the contribution of each view and the global information, respectively. By jointly optimizing these components, the proposed method integrates the complementary strengths of individual views and global data patterns, ensuring robust clustering results.

The complementary information between views is explicitly captured by minimizing the term $\sum_v W_v \|S - A^{(v)}\|_F^2$. This enforces alignment between the Global Similarity Matrix $S$ and individual view-specific matrices $A^{(v)}$, ensuring that $S$ incorporates the unique contributions of each view. At the same time, the term $\beta W \|S - A\|_F^2$ promotes consistency with the Global Affinity Matrix $A$, which represents an aggregated perspective of all views. By balancing these terms, the method ensures that $S$ reflects both shared and unique characteristics of the data.

## Optimization

We developed the following optimization scheme to solve our objective function (9). For convenience, we present the objective function (9) in the following form.

$$\min_{S,H} \sum_{v=1}^{m} \sqrt{\sum_{ij} (a_{ij}^{(v)} - s_{ij})^2} + \beta \sqrt{\sum_{ij} (a_{ij} - s_{ij})^2} + \lambda \text{Tr}(H^T L_S H)$$
$$\text{s.t.} \quad \sum_{j} s_{ij} = 1, s_{ij} \geq 0, H \in \mathbb{R}^{n \times c}, H^T H = I. \tag{11}$$

Meanwhile, the objective function (9) with the rank constraint is difficult to solve. Based on Theorems 1 and 2, we relaxed it by replacing the rank constraint with the minimization of $\text{Tr}(H^T L_S H)$ for easier optimization.

Inspired by SwMC, we learn the weight parameters for each view and the global view adaptively.

$$W_v = \frac{1}{2\sqrt{\sum_{ij} (a_{ij}^{(v)} - s_{ij})^2}}, \quad W = \frac{1}{2\sqrt{\sum_{ij} (a_{ij} - s_{ij})^2}} \tag{12}$$

Then problem (11) becomes:

$$\min_{S,H} \sum_{v=1}^{m} W_v \sum_{ij} (a_{ij}^{(v)} - s_{ij})^2 + \beta W \sum_{ij} (a_{ij} - s_{ij})^2 + \lambda \text{Tr}(H^T L_S H)$$
$$\text{s.t.} \quad \sum_{j} s_{ij} = 1, \quad s_{ij} \geq 0, \quad H \in \mathbb{R}^{n \times c}, \quad H^T H = I. \tag{13}$$

For problem (13), we can solve $S$, $W$, and $W_v$ iteratively.

**(1) Fix $W$ and $W_v$, update $S$:**

**i. Solving $H$ when $S$ is fixed:**

When $S$ is fixed, problem (12) becomes:

$$\min_{H \in \mathbb{R}^{n \times c}, H^T H = I} \mathrm{Tr}(H^T L_S H) \tag{14}$$

It is known that the optimal solution for $H$ consists of the $c$ eigenvectors of $L_S \in \mathbb{R}^{n \times n}$ corresponding to the smallest $c$ eigenvalues. Since $c \ll n$, problem (13) can be efficiently solved using the Arnoldi iteration method.

**ii. Solving $S$ when $H$ is fixed:**

When $H$ is fixed, problem (12) becomes:

$$\min_{s_{ij} \geq 0, \sum_j s_{ij} = 1} \sum_v W_v \sum_{ij} (a_{ij}^{(v)} - s_{ij})^2 + \beta W \sum_{ij} (a_{ij} - s_{ij})^2 + \frac{1}{2\lambda} \sum_{ij} \|h_i - h_j\|_2^2 s_{ij} \tag{15}$$

Since problem (12) is independent for different $i$, we can solve the following problem for each $i$ separately:

$$\min_{s_{ij} \geq 0, \sum_j s_{ij} = 1} \sum_v W_v \sum_i (a_{ij}^{(v)} - s_{ij})^2 + \beta W \sum_i (a_{ij} - s_{ij})^2 + \frac{1}{2\lambda} \sum_i \|h_i - h_j\|_2^2 s_{ij} \tag{16}$$

For simplicity, let $\eta_{ij} = \|h_i - h_j\|_2^2$, and let $\eta_i$ be a vector whose $j$th element is equal to $\eta_{ij}$. Since the $j$th element of $a_i$ equals $a_{ij}$, and the $j$th element of $a_i^{(v)}$ equals $a_{ij}^{(v)}$, problem (15) can be written in the following vector form:

$$\min_{s_i \cdot \mathbf{1}_n = 1, s_i \geq 0} \left\| s_i - \frac{\sum_{v=1}^m W_v a_i^{(v)} + \beta W a_i - \frac{1}{4}\lambda \eta_i}{\sum_v W_v + \beta W} \right\|_2^2 \tag{17}$$

This problem is a constrained convex optimization problem. The objective function is the squared L2-norm of $s_i$, which is a convex function. The constraints include $s_i \cdot \mathbf{1}_n = 1$, which is a linear constraint requiring the sum of all elements to equal 1, and $s_i \geq 0$, which is an element-wise non-negativity constraint.

Linear constraints and non-negativity constraints are both convex sets, so this is a typical convex optimization problem with a convex set as the constraint space. It can be solved using classical methods such as the projected gradient method, where the vector is projected back onto the constraint space during gradient descent. For this type of problem, when the matrix to be updated in each iteration of the projected gradient method is sparse, meaning that only $k$ vectors in the entire linear space need to be updated at each iteration, the "sparse projected gradient method" proposed by Duchi et al. [25]. Duchi et al. provides an efficient solution, reducing the complexity of each iteration to $O(k \log(n))$. Referring to the case of a positive simplex as the constraint space in [25], to accelerate the solution of $S$, in each iteration, we choose to update only the $k$ neighbors of data $i$. We set $k$ to a small constant to ensure that $S$ is completely sparse, then use the sparse projected gradient method to quickly complete an iteration.

**(2) Fix $S$, update $W$ and $W_v$:**

Based on the current $S$, use equation (11) to calculate the current $W$ and $W_v$.

The above optimization scheme can be expressed as the following pseudocode (Algorithm 1):

**Algorithm 1. Solve the objective function.**

1: **Input:** Initial affinity matrices $A^{(v)} \in \mathbb{R}^{n \times n}$ from the $v$-th view, global affinity matrix $A \in \mathbb{R}^{n \times n}$, number of clusters $c$, number of views $m$.

2: Set $W_v = W = 1/(m+1)$.

3: Initialize $S = (\sum_v A^{(v)} + \beta A)/(m+1)$.

4: Compute $H \in \mathbb{R}^{n \times c}$, which is formed by the $c$ smallest eigenvectors of $L_S$.

5: **while** not converged **do**

6: **for** each $i \in \{1, \ldots, n\}$ **do**

7: Update the $i$th row of $S$ by solving problem (17).

8: **end for**

9: **end while**

10: Update $W_v$ and $W$ using Eq.(12).

11: **Output:** The matrix $S \in \mathbb{R}^{n \times n}$ with exactly $c$ connected components.

## Convergence analysis

**Lemma 1.** Nie et al. [26] proved that for any positive numbers $u$ and $v$, the following inequality holds:

$$u - \frac{u^2}{2v} \leq v - \frac{v^2}{2v}. \tag{18}$$

Based on Lemma 1, we can prove that the Algorithm 1 will monotonically decrease the objective of Eq.(9) in each iteration. The proof is as follows.

**Proof:** According to Algorithm 1, we have

$$\sum_{v=1}^{m} \frac{\|S_{t+1} - A^{(v)}\|_F^2}{2\|S_t - A^{(v)}\|_F} + \beta \frac{\|S_{t+1} - A\|_F^2}{2\|S_t - A\|_F} \leq \sum_{v=1}^{m} \frac{\|S_t - A^{(v)}\|_F^2}{2\|S_t - A^{(v)}\|_F} + \beta \frac{\|S_t - A\|_F^2}{2\|S_t - A\|_F}. \tag{19}$$

where $t$ and $t+1$ denote the $t$-th and $(t+1)$-th iterations, respectively.

According to Lemma 1, we have

$$\|S_{t+1} - A^{(v)}\|_F - \frac{\|S_{t+1} - A^{(v)}\|_F^2}{2\|S_t - A^{(v)}\|_F} \leq \|S_t - A^{(v)}\|_F - \frac{\|S_t - A^{(v)}\|_F^2}{2\|S_t - A^{(v)}\|_F} \tag{20}$$

$$\beta \left( \|S_{t+1} - A\|_F - \frac{\|S_{t+1} - A\|_F^2}{2\|S_t - A\|_F} \right) \leq \beta \left( |S_t - A|_F - \frac{\|S_t - A\|_F^2}{2\|S_t - A\|_F} \right) \tag{21}$$

Thus, for all views, we have

$$\sum_{v=1}^{m} \|S_{t+1} - A^{(v)}\|_F - \frac{\|S_{t+1} - A^{(v)}\|_F^2}{2\|S_t - A^{(v)}\|_F} + \beta \left( \|S_{t+1} - A\|_F - \frac{\|S_{t+1} - A\|_F^2}{2\|S_t - A\|_F} \right)$$

$$\leq \tag{22}$$

$$\sum_{v=1}^{m} \|S_t - A^{(v)}\|_F - \frac{\|S_t - A^{(v)}\|_F^2}{2\|S_t - A^{(v)}\|_F} + \beta \left( |S_t - A|_F - \frac{\|S_t - A\|_F^2}{2\|S_t - A\|_F} \right)$$

By simple algebra, we obtain

$$\sum_{v=1}^{m}\|S_{t+1}-A^{(v)}\|_F+\beta\|S_{t+1}-A\|_F-\left(\sum_{v=1}^{m}\frac{\|S_{t+1}-A^{(v)}\|_F^2}{2\|S_t-A^{(v)}\|_F}+\beta\frac{\|S_{t+1}-A\|_F^2}{2\|S_t-A\|_F}\right)$$
$$\leq \tag{23}$$
$$\sum_{v=1}^{m}\|S_t-A^{(v)}\|_F+\beta|S_t-A\|_F-\left(\sum_{v=1}^{m}\frac{\|S_t-A^{(v)}\|_F^2}{2\|S_t-A^{(v)}\|_F}+\beta\frac{\|S_t-A\|_F^2}{2\|S_t-A\|_F}\right)$$

By combining Eq. (19) and Eq. (23)

$$\sum_{v=1}^{m}\|S_{t+1}-A^{(v)}\|_F+\beta\|S_{t+1}-A\|_F\leq\sum_{v=1}^{m}\|S_t-A^{(v)}\|_F+\beta|S_t-A\|_F \tag{24}$$

Thus, the Algorithm 1 will monotonically decrease the objective of Eq.(9) in each iteration.

## Complexity analysis

The computational complexity of the proposed algorithm is primarily determined by the solutions for four variables: $S$, $H$, $W$, and $W_v$. Given that $m\ll n$, $m$ is neglected in the analysis.. The solution for $S$ uses the sparse projected gradient method based on [25]. Under the condition that only the $k$ neighbors of data $i$ are updated in each iteration, the complexity is $O(k\cdot\log(n))$. The solution for $H$ uses the Arnoldi iteration method, with a complexity of approximately $O(n^2\cdot c)$. The computational complexity for $W$ and $W_v$ is $O(n^2)$, but since only addition and subtraction are involved in each calculation, the actual computation is very fast. In summary, given that $c\ll n$, the computational complexity of our algorithm in one iteration is approximately $O(n^2)+O(k\cdot\log(n))$.

## Experimental results and analysis

### Hardware setup and execution time

The experiments were conducted on a machine equipped with an **Intel Xeon Platinum 8352V CPU (2 processors, 2.10GHz base frequency, 3.50GHz turbo frequency)** and **128GB RAM**. No GPU was used in the experiments, demonstrating the efficiency of our method even in a CPU-only environment. Table 1 summarizes the execution times for each dataset. The results demonstrate that our method is efficient and suitable for deployment on resource-limited devices.

### Convergence behavior

To visually demonstrate the convergence behavior of the proposed algorithm, we plotted the convergence curves for the optimization process on the ORL and Yale datasets. Fig 1 illustrates how the objective function value changes over the iterations.

**Table 1. Execution times for each dataset.**

| Dataset | Number of Samples | Execution Time (seconds) |
|---|---|---|
| ORL | 400 | 0.9187 |
| Yale | 165 | 0.3340 |

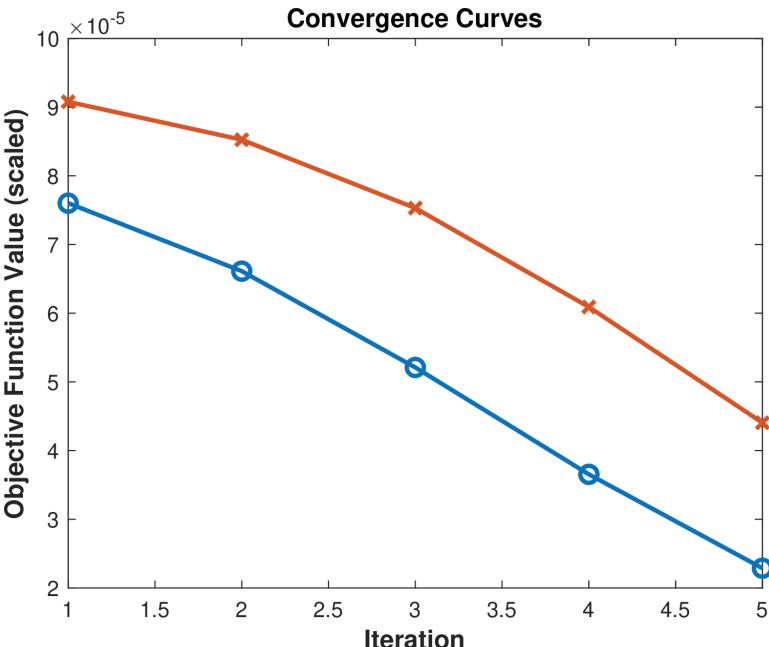

**Fig 1. Convergence curves of the proposed algorithm on the ORL and Yale datasets.** The objective function value decreases rapidly in the first few iterations and stabilizes as the algorithm converges.

The curves show that the algorithm converges within a small number of iterations, indicating its efficiency in solving the optimization problem. The convergence curves are consistent across datasets, demonstrating the robustness of our method.

## Visualization of clustering results

To provide an intuitive understanding of the clustering performance, we visualized the extracted features for the Yale and ORL datasets using t-SNE.

Fig 2 demonstrates the clustering results for the Yale (a) and ORL (b) datasets. Each point represents an instance, and the colors indicate the ground-truth cluster assignments. It can be observed that the extracted features group naturally into distinct clusters, demonstrating the effectiveness of our clustering algorithm. For both datasets, the majority of clusters are well-separated, indicating good clustering performance.

## Performance comparison

In this section, we evaluate and compare the performance of the proposed method on four widely used multi-view datasets:

The MSRC-v1 dataset [27] consists of 240 images in eight categories. Following [28], we selected 7 categories with 30 images each and extracted five visual features: 24-D color moments, 512-D GIST, 576-D HOG, 254-D CENTRIST, and 256-D LBP.

The Caltech101 dataset [29] includes 101 categories. We used 441 samples from 7 categories, constructing three views: 2560-D SIFT, 1160-D LBP, and 620-D HOG.

The ORL dataset contains 400 facial images of 40 individuals under varying conditions. We constructed three views: 6750-D Gabor, 4096-D intensity, and 3340-D LBP features.

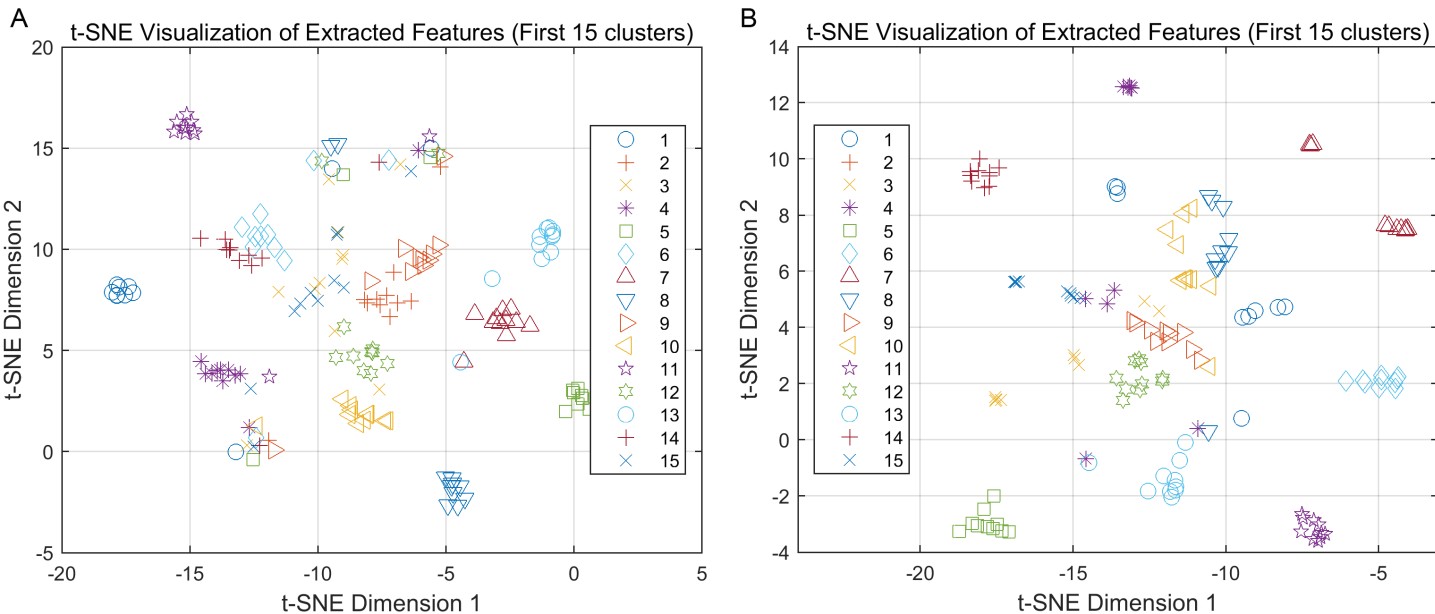

**Fig 2. Visualization of clustering results on Yale and ORL datasets.** (a) Clustering visualization for the Yale dataset using t-SNE, showing distinct clusters corresponding to extracted features. (b) Clustering visualization for the ORL dataset using t-SNE, illustrating clear separation between clusters.

The Yale dataset has 165 images of 15 individuals under different conditions. We constructed three views: 3304-D LBP, 6750-D Gabor, and 4096-D intensity features.

We compare our method with the following five multi-view clustering algorithms:

1. (CSMSC): A subspace learning method that captures the consistency information of multiple views;
2. Auto-weighted Multi-Graph Learning (AMGL) [18]: Constructs graphs for each single view and automatically learns the optimal weight for each graph;
3. Robust Multi-view Spectral Clustering (RMSC) [14]: Recovers a latent transition probability matrix from the matrices computed from each single view for multi-view clustering;
4. Multi-view Clustering based on Consensus Matrix Learning (MCGC) [16];
5. Multi-view Clustering based on Adaptive Graph Learning (MVGL) [17].

To evaluate performance, we use Accuracy (ACC), Normalized Mutual Information (NMI), and Purity as clustering performance measures. Higher values of these three metrics indicate better performance.

For each comparison method, we adjusted the parameters to achieve optimal results. Since the performance of K-means clustering is highly sensitive to the choice of initial centroids, all methods involving K-means are repeated 10 times, and the average results are reported. As for our MCGGL, we run it only once.

Table 2 lists the ACC, NMI, and Purity values of the seven methods on the four datasets mentioned above, where SC-best refers to performing spectral clustering on each individual view and selecting the best result.

From observing Table 2, we can draw the following conclusions:

**Table 2. The clustering performances on Caltech101, MSRC-V1, Yale, and ORL datasets.**

| Dataset | MSRC-V1 | | | Caltech101 | | |
|---|---|---|---|---|---|---|
| Metric | ACC | NMI | Purity | ACC | NMI | Purity |
| SC-best | 0.663 | 0.534 | 0.674 | 0.545 | 0.431 | 0.624 |
| CSMSC | 0.741 | 0.598 | 0.733 | 0.563 | 0.472 | 0.628 |
| AMGL | 0.732 | 0.665 | 0.738 | 0.479 | 0.342 | 0.539 |
| RMSC | 0.742 | 0.638 | 0.763 | 0.525 | 0.289 | 0.561 |
| MCGC | 0.852 | 0.724 | 0.852 | 0.501 | 0.373 | 0.586 |
| MVGL | 0.843 | 0.744 | 0.841 | 0.482 | 0.370 | 0.565 |
| Ours | **0.884** | **0.797** | **0.898** | **0.674** | **0.681** | **0.732** |
| Dataset | ORL | | | Yale | | |
| Metric | ACC | NMI | Purity | ACC | NMI | Purity |
| SC-best | 0.727 | 0.868 | 0.762 | 0.556 | 0.586 | 0.567 |
| CSMSC | 0.853 | 0.926 | 0.873 | 0.743 | 0.775 | 0.748 |
| AMGL | 0.767 | 0.881 | 0.816 | 0.641 | 0.651 | 0.653 |
| RMSC | 0.743 | 0.857 | 0.755 | 0.701 | 0.716 | 0.702 |
| MCGC | 0.781 | 0.882 | 0.814 | 0.713 | 0.672 | 0.664 |
| MVGL | 0.763 | 0.870 | 0.812 | 0.705 | 0.689 | 0.704 |
| Ours | **0.906** | **0.931** | **0.910** | **0.857** | **0.849** | **0.831** |

- Multi-view clustering methods generally outperform **SC-best** and achieve the best performance. The reason may be that multi-view representations provide more information compared to single-view representations. By utilizing more information, multi-view methods achieve better clustering results.

- CLR-based multi-view clustering methods, such as **MCGC** and our proposed method, outperform most other methods on most datasets. This may be because CLR-based clustering methods directly obtain the clustering labels of the data, whereas other methods rely on the quality of the input graphs and require additional post-processing steps, such as K-means, which can lead to suboptimal solutions.

- Our proposed method outperforms other clustering methods. This may be because our method, during the Global-view Graph learning process, effectively captures the specificity of each view while also considering the complementary information between different views, thus fully integrating the complementary information and global information of the entire dataset.

- Our method achieves relatively good clustering accuracy across multiple multi-view datasets, including Caltech101, MSRC-v1, Yale, and ORL. This consistent high performance demonstrates the strong generalization ability of our model, making it adaptable to the characteristics of different datasets and capable of producing robust clustering results.

## Parameter analysis

In this section, we analyze the effect of the parameter $\beta$ on our algorithm. We adjusted the parameter $\beta$ to values $[0, 0.01, 0.1, 0.2, 0.3, 0.4, 0.5, 1, 10]$, conducted ten experiments, and compared the results. Similarly, we chose ACC, NMI, and Purity as evaluation metrics.

Fig 3 shows the clustering performance as $\beta$ varies across four datasets. Without parameter tuning, the method cannot fully capture complementary and global information from multiple views. From Fig 3, we can see that as $\beta$ changes, our method shows significant fluctuations, and when $\beta = 0$, our method performs worse across all four datasets compared to the best performance when $\beta = 0.3$. This also indicates that Global-view Graph Learning helps improve clustering performance.

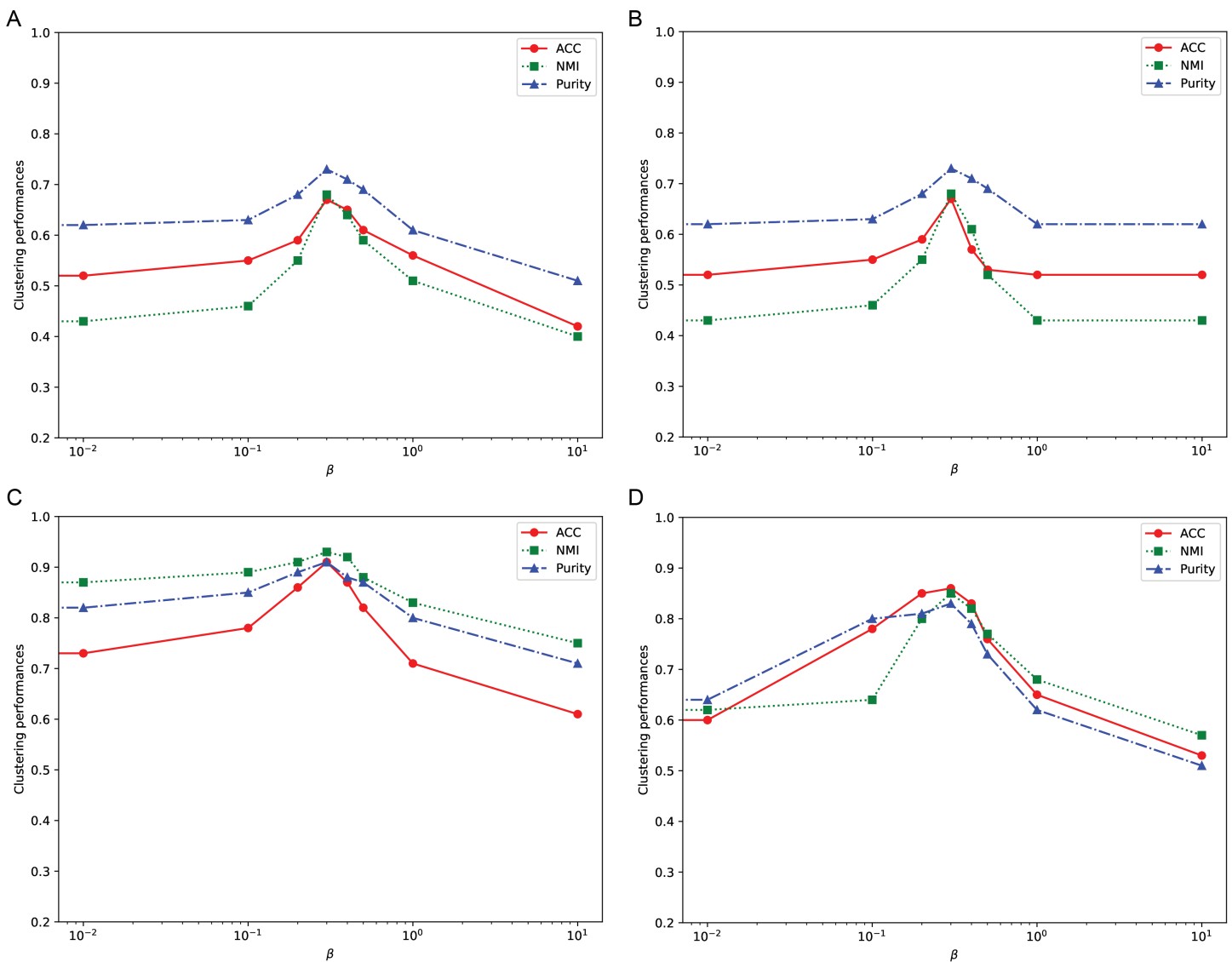

**Fig 3. The effect of different values of $\beta$ on clustering metrics with different datasets.** (a) MSRC-V1, (b) Caltech101, (c) ORL, and (d) Yale. The figure illustrates how the clustering metrics change as the value of $\beta$ varies.

## Discussion

Our method effectively integrates complementary information between different views and the global information of the entire dataset. This integration ensures superior performance compared to various spectral clustering methods across multiple datasets, as demonstrated by consistent improvements in clustering metrics such as ACC, NMI, and Purity.By constructing a Global-view Graph, our approach captures both the specificity of each view and the complementary information between views, which leads to robust clustering results.

In terms of computational efficiency, our method leverages the CLR-based framework to achieve low computational complexity. Solving the proposed optimization objective results in a computational complexity of approximately $O(n^2 + O(k \cdot \log(n)))$ per iteration, making it more suitable for large-scale clustering tasks than many existing approaches.

Spectral clustering methods based on tensor analysis, in contrast, typically have a higher computational complexity of $O(n^3)$. For instance, the computational complexity of a single iteration in Xia et al. [21] is approximately $O(n^3 + n^2 \cdot d + n^2 \cdot \log(n))$. While tensor analysis-based methods perform well in terms of accuracy, their high computational cost limits their practical application in resource-constrained environments.

Clustering networks based on deep learning also exhibit high computational complexity. For example, consider a deep autoencoder network with depth $L$. Assuming $n$ samples aggregated into $c$ clusters, and with both the sample dimension and hidden layer feature dimension as $d$, a single iteration involves the following operations:

(1) Forward propagation through each layer of the encoder network with a complexity of $O(n \cdot d^2 \cdot L)$.
(2) Backpropagation with a similar complexity of $O(n \cdot d^2 \cdot L)$.
(3) Updating the clustering loss, assuming k-means is used, with a complexity of $O(n \cdot c \cdot d)$.

The total complexity is $O(n \cdot d^2 \cdot L + n \cdot c \cdot d)$. Given that $d^2 > n$, this complexity is often higher than the square of the number of samples and grows rapidly with the depth of the network.

If large models like BERT [30] are used for clustering tasks, the computational complexity increases further due to the Transformer architecture [31]. Assuming the self-attention layer has $h$ heads, the main operations involved in a single iteration include:

(1) Dense matrix multiplication in the self-attention layer during forward propagation, with a complexity of $O(n \cdot d^3 \cdot L \cdot h)$.
(2) Dense matrix multiplication in the feedforward network (FFN) during forward propagation, with a complexity of $O(n \cdot d^3 \cdot L)$.
(3) Backpropagation, which has a complexity similar to forward propagation, $O(n \cdot d^3 \cdot L \cdot h)$.
(4) Updating the clustering loss, with a complexity of $O(n \cdot c \cdot d)$.

The total complexity of a single iteration of BERT is approximately $O(n \cdot d^3 \cdot L \cdot h + n \cdot c \cdot d)$. This highlights that deep learning-based approaches, especially those using large models, impose significant computational demands.

In summary, our method achieves a favorable balance between performance and computational efficiency. By reducing algorithmic complexity while maintaining high clustering accuracy, our approach is well-suited for deployment in resource-constrained environments. Potential applications include devices such as mobile phones, tablets, drones, and edge computing gateways. Furthermore, our method's efficiency and performance make it a strong candidate for large-scale real-world clustering tasks, where computational resources may be limited.

While the proposed algorithm demonstrates superior performance and efficiency, we acknowledge its sensitivity to the parameter $\beta$, which balances the contribution of global and view-specific information during clustering. As shown in our parameter analysis section, the choice of $\beta$ significantly impacts the clustering results. Improper tuning of $\beta$ can lead to suboptimal performance, particularly for datasets with high variability in view-specific or global information. To mitigate this issue, future work could explore automatic parameter selection methods or parameter-free approaches to enhance the robustness of the algorithm.

## Conclusion and future work

In this paper, we propose a CLR-based multi-view clustering method to learn a Global-view Graph with exactly $c$ connected components, which is an ideal structure for clustering. The

proposed Multi-view Clustering via Global-view Graph Learning (MCGGL) aims to extract the specific information of each view while considering the complementary information between different views, thereby fully integrating the complementary information between different views and the global information of the overall data. To achieve this goal, we construct an objective function and derive a simple and effective optimization algorithm to solve this objective function, resulting in low computational complexity for our algorithm. Our method demonstrates significant improvements in clustering performance across diverse datasets by fully integrating complementary and global information from multiple views. Experimental results show that the clustering performance of our proposed method is superior to other clustering methods.

In the future, we will attempt to integrate CLR-based, tensor analysis-based, and deep learning-based multi-view clustering methods. For example, we will introduce the idea of directly assigning clustering labels to data from CLR into tensor analysis-based multi-view clustering methods to ensure that the model achieves an optimal solution. Furthermore, we can attempt to construct a tensor analysis neural network connected to a deep auto-encoder network, creating an end-to-end tensor analysis-based multi-view clustering network. This approach would fully leverage the advantages of tensor analysis in multi-view data fusion, increase the interpret-ability of deep learning networks, and effectively control the network's width and depth, thereby improving clustering performance while reducing computational resource consumption.

## Author contributions

**Conceptualization:** Qin Li.

**Formal analysis:** Geng Yang.

**Funding acquisition:** Qin Li.

**Methodology:** Qin Li.

**Validation:** Geng Yang.

**Writing – original draft:** Qin Li.

**Writing – review & editing:** Geng Yang.

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
