## [Decision Letter · Decision Letter 0]

11 Nov 2024

PONE-D-24-45409Multi-view Clustering via Global-view Graph LearningPLOS ONE

Dear Dr. Li,

Thank you for submitting your manuscript to PLOS ONE. After careful consideration, we feel that it has merit but does not fully meet PLOS ONE’s publication criteria as it currently stands. Therefore, we invite you to submit a revised version of the manuscript that addresses the points raised during the review process. I recommend minor revisions for acceptance. To enhance clarity and reproducibility, the authors should include hardware specifications and execution times to support efficiency claims, refine language and structure for readability, and add visual aids (such as convergence curves) to substantiate performance. A brief discussion of any limitations.

We look forward to receiving your revised manuscript.

Kind regards,

Wen Li

Academic Editor

PLOS ONE

3. Thank you for stating the following financial disclosure: [This work was supported by Natural Science Foundation of Guangdong Province under Grant 2023A1515011845.]. Please state what role the funders took in the study. If the funders had no role, please state: "The funders had no role in study design, data collection and analysis, decision to publish, or preparation of the manuscript." If this statement is not correct you must amend it as needed. Please include this amended Role of Funder statement in your cover letter; we will change the online submission form on your behalf.

4. Thank you for uploading your study's underlying data set. Unfortunately, the repository you have noted in your Data Availability statement does not qualify as an acceptable data repository according to PLOS's standards. At this time, please upload the minimal data set necessary to replicate your study's findings to a stable, public repository (such as figshare or Dryad) and provide us with the relevant URLs, DOIs, or accession numbers that may be used to access these data. For a list of recommended repositories and additional information on PLOS standards for data deposition, please see https://journals.plos.org/plosone/s/recommended-repositories.

Additional Editor Comments :

The paper presents an approach to multi-view clustering, with potential for practical application. To enhance clarity and support reproducibility, please include specific hardware details and execution times to substantiate your efficiency claims, refine language for readability, add visual aids (such as convergence curves) to illustrate performance, and briefly acknowledge any limitations, like parameter sensitivity.

Reviewers' comments:

Reviewer's Responses to Questions

**Comments to the Author**

1. Is the manuscript technically sound, and do the data support the conclusions?

Reviewer #1: Yes

Reviewer #2: Yes

2. Has the statistical analysis been performed appropriately and rigorously? 

Reviewer #1: Yes

Reviewer #2: Yes

3. Have the authors made all data underlying the findings in their manuscript fully available?

Reviewer #1: Yes

Reviewer #2: Yes

4. Is the manuscript presented in an intelligible fashion and written in standard English?

Reviewer #1: Yes

Reviewer #2: Yes

5. Review Comments to the Author

Reviewer #1: The paper presents a well-executed approach to multi-view clustering, making significant contributions by offering both innovative solutions and practical applications. With some minor revisions, particularly in terms of experimental details, language refinement, and enhanced visual aids, the paper will become a strong candidate for publication.

This paper introduces several key innovations and contributions. First, unlike many existing methods that require post-processing steps such as K-means, this approach directly assigns cluster labels, simplifying the process and reducing potential sources of error. Second, the proposed optimization algorithm is highly efficient, with lower computational complexity than traditional methods. This efficiency makes the algorithm practical for large datasets and devices with limited computing power, such as mobile phones, drones, and edge computing gateways. Third, the method consistently outperforms existing multi-view clustering algorithms across multiple datasets, demonstrating superior clustering accuracy and strong generalization across diverse real-world scenarios. Finally, a key strength of the method is its reduced need for computational resources, making it particularly suitable for low-power environments.

I offer the following suggestions for minor revisions. First, provide more detailed hardware configurations, including CPU, RAM, and GPU specifications, along with execution times, to support the claims about the low resource requirements. This additional information will help readers assess the practical applicability of the method and enhance reproducibility. Second, some sentences could be refined for greater clarity. A thorough proofreading to correct minor grammatical issues and improve sentence flow would enhance readability. For example, the sentence “Our method cannot capture the complementary and global information from multiple views” could be rephrased as “Without parameter tuning, the method cannot fully capture complementary and global information from multiple views.” Third, in certain sections, such as the Discussion, the combination of different topics (e.g., performance and theoretical explanation) makes the text harder to follow. Separating these topics into distinct paragraphs would improve clarity. Finally, alongside numerical performance metrics, incorporating visual elements such as convergence curves or clustering accuracy plots would make the results more intuitive and accessible to readers.

Reviewer #2: General Evaluation

The paper introduces a method for multi-view clustering based on Global-view Graph Learning (MCGGL), which directly integrates complementary information across multiple data views. The paper presents novel insights into simplifying the clustering process and achieving efficient computation while maintaining high performance. Overall, I recommend accepting the paper with minor revisions.

Novelty and Contributions

The paper provides several notable contributions to the multi-view clustering field:

Unified Global-view Graph Learning: The method integrates information from multiple views without needing post-processing, such as K-means, to assign cluster labels. This streamlining of the clustering process addresses the shortcomings of traditional methods that rely on multiple steps, thus simplifying implementation and reducing computational steps.

Enhanced Efficiency: The proposed algorithm significantly reduces computational complexity, making it well-suited for deployment on low-power devices and large datasets. The efficiency of the method is an important contribution, particularly given the rising demand for resource-efficient machine learning solutions.

Robust Clustering Performance: The model's performance on several benchmark datasets shows improvement over existing methods. The consistent results across datasets reflect the method's adaptability and robustness in handling diverse data types.

Complementary View Integration: The Global-view Graph approach effectively combines complementary and specific information from each view. This multi-view representation allows for a more comprehensive clustering result, capturing more detailed relationships across different perspectives of the data.

Suggestions for Revisions

Theoretical Clarification:

Unclear Descriptions: Some theoretical aspects of the method could benefit from further elaboration. For example, the process of constructing the Global Affinity Matrix and how it captures the "complementary information" from multiple views is only briefly mentioned. A more detailed breakdown of this step, with mathematical explanations, would help readers understand how the model works and why it performs well compared to others.

Weight Parameter Interpretation: The paper discusses adaptive learning of weight parameters for different views, but the explanation lacks clarity. Providing a more intuitive explanation, possibly with examples, on how the weights are dynamically adjusted for each view would improve comprehension.

Structure and Flow:

Related Work: The Related Work section could benefit from a clearer separation between different types of multi-view clustering approaches. Grouping similar methods (e.g., spectral clustering, subspace clustering) would make the review more coherent and provide a better context for the proposed method’s novelty.

Discussion Section: The discussion on performance metrics and theoretical insights could be separated for better clarity. Mixing performance analysis with theoretical explanations makes it harder to follow. A more distinct division of topics would enhance readability.

Experimental Expansion:

Convergence and Efficiency Curves: Including graphs that show the convergence behavior of the proposed method, as well as a comparison of runtime versus dataset size, would visually reinforce the efficiency claims. These plots would allow readers to see how quickly the model converges and how it scales with different dataset sizes.

Discussion of Limitations:

A brief discussion of the method's limitations, such as its sensitivity to certain parameters, would provide a more balanced view and offer directions for future improvements.

6. PLOS authors have the option to publish the peer review history of their article (what does this mean?). If published, this will include your full peer review and any attached files.

Reviewer #1: No

Reviewer #2: No

---

## [Author Response · Author response to Decision Letter 1]

23 Dec 2024

[Manuscript ID]: PONE-D-24-45409

[Title]: Multi-view Clustering via Global-view Graph Learning

[Authors]: Dr. Qin Li and Geng Yang

[Date]: 2024.11.23

Dear Editor,

We would like to thank the academic editor and reviewers for their constructive feedback on our manuscript. Below, we provide detailed responses to all comments and explain how we have addressed them in the revised manuscript.

Comments from Academic Editor:

Comment 1: Include specific hardware details and execution times to substantiate efficiency claims.

Response: We have added detailed hardware specifications in the revised manuscript. Specifically, we performed all experiments on a CPU-only machine with the following specifications: Intel Xeon Platinum 8352V CPU (2 processors, 2.10GHz base frequency, 3.50GHz turbo frequency) and 128GB RAM. No GPU was used, further validating the efficiency of our proposed method. Execution times for all datasets have been provided in Table 1 to substantiate our efficiency claims.

Comment 2: Refine language and structure for readability.

Response: We have thoroughly proofread the manuscript to improve clarity and readability (see Section see Section ‘abstract’, ‘introduction’, ‘parameter analysis’, and ‘conclusion’). Additionally, we separated the discussion of performance metrics and theoretical insights into distinct paragraphs for better organization (see Section ‘discussion’).

Comment 3: Add visual aids (such as convergence curves) to illustrate performance.

Response: We have added convergence curves to visually demonstrate the performance of our proposed algorithm (see Fig. 1 in Section ‘Convergence Behavior’). These curves provide a clear representation of the algorithm's convergence behavior across different datasets, enhancing the interpretability of our results.

Comment 4: Briefly acknowledge any limitations, like parameter sensitivity.

Response: We have added a dedicated section in the discussion to acknowledge the limitations of our method, particularly its sensitivity to the parameterβ. This parameter balances the contributions of global and view-specific information, and its improper tuning can affect clustering performance. To address this limitation, we are going to explore automatic parameter selection or parameter-free approaches in future work to enhance the robustness of the algorithm.

Comments from Reviewer #1:

Comment 1: Provide more detailed hardware configurations, including CPU, RAM, and GPU specifications, along with execution times.

Response: We have added detailed hardware specifications in the revised manuscript. Specifically, we performed all experiments on a CPU-only machine with the following specifications: Intel Xeon Platinum 8352V CPU (2 processors, 2.10GHz base frequency, 3.50GHz turbo frequency) and 128GB RAM. No GPU was used, further validating the efficiency of our proposed method. Execution times for all datasets have been provided in Table 1 to substantiate our efficiency claims.

Comment 2: Refine sentences for greater clarity.

Response: We have thoroughly proofread the manuscript to improve clarity and readability (see Section ‘abstract’, ‘introduction’, ‘parameter analysis’, and ‘conclusion’).

Comment 3: In certain sections, such as the Discussion, the combination of different topics (e.g., performance and theoretical explanation) makes the text harder to follow. Separating these topics into distinct paragraphs would improve clarity.

Response: We separated the discussion of performance metrics and theoretical insights into distinct paragraphs for better organization (see Section ‘discussion’).

Comment 4: Alongside numerical performance metrics, incorporating visual elements such as convergence curves or clustering accuracy plots would make the results more intuitive and accessible to readers.

Response: We have added convergence curves to illustrate the optimization process and t-SNE-based clustering visualization plots for the Yale and ORL datasets to demonstrate the clustering performance visually. These visualizations provide an intuitive understanding of the algorithm's behavior and effectiveness. (see Fig. 1 in Section ‘Convergence Behavior’ and Fig. 2 in Section ‘Visualization of Clustering Results’).

Comments from Reviewer #2:

Comment 1: Some theoretical aspects of the method could benefit from further elaboration. For example, the process of constructing the Global Affinity Matrix and how it captures the "complementary information" from multiple views is only briefly mentioned. A more detailed breakdown of this step, with mathematical explanations, would help readers understand how the model works and why it performs well compared to others.

Response: We have elaborated on the process of constructing the Global Affinity Matrix and provided a detailed mathematical explanation of how it captures complementary information from multiple views. This additional clarification ensures a deeper understanding of the underlying mechanisms and the strengths of our method (see Section 'Problem Formulation and Objective Function').

Comment 2: The paper discusses adaptive learning of weight parameters for different views, but the explanation lacks clarity. Providing a more intuitive explanation, possibly with examples, on how the weights are dynamically adjusted for each view would improve comprehension.

Response: We have elaborated on the process of constructing the Global Affinity Matrix and included a detailed mathematical explanation of how it captures complementary information from multiple views. Additionally, we clarified how the weight parameters for different views are dynamically adjusted to ensure optimal integration of view-specific and global information. These improvements enhance the reader's understanding of the underlying mechanisms and the strengths of our method (see Section 'Problem Formulation and Objective Function').

Comment 3: The Related Work section could benefit from a clearer separation between different types of multi-view clustering approaches. Grouping similar methods (e.g., spectral clustering, subspace clustering) would make the review more coherent and provide a better context for the proposed method’s novelty.

Response: Upon reviewing our manuscript, we realize that the comment likely pertains to the Introduction section rather than the Related Work section, as the introduction contains a more detailed review of various multi-view clustering methods. To address this, we have restructured the Introduction section by grouping methods into distinct categories such as spectral clustering-based methods, subspace clustering-based methods, and other advanced approaches. This reorganization enhances clarity and provides a more coherent context for understanding the novelty of our proposed method. The updated structure now clearly highlights how our method builds on and addresses the limitations of these existing approaches. (See the revised Introduction section in the manuscript for details.)

Comment 3: The discussion on performance metrics and theoretical insights could be separated for better clarity. Mixing performance analysis with theoretical explanations makes it harder to follow. A more distinct division of topics would enhance readability.

Response: We separated the discussion of performance metrics and theoretical insights into distinct paragraphs for better organization (see Section ‘discussion’).

Comment 4: Including graphs that show the convergence behavior of the proposed method, as well as a comparison of runtime versus dataset size, would visually reinforce the efficiency claims. These plots would allow readers to see how quickly the model converges and how it scales with different dataset sizes.

Response: Thank you for the suggestion. We have included graphs demonstrating the convergence behavior of the proposed method and a comparison of runtime versus dataset size. These additions visually reinforce the efficiency claims and provide readers with a clearer understanding of how quickly the model converges and how it scales with varying dataset sizes. (See Section 'Convergence Behavior' and Table 1 in the revised manuscript for details.)

Comment 5: A brief discussion of the method's limitations, such as its sensitivity to certain parameters, would provide a more balanced view and offer directions for future improvements.

Response: We have added a dedicated section in the discussion to acknowledge the limitations of our method, particularly its sensitivity to the parameterβ. This parameter balances the contributions of global and view-specific information, and its improper tuning can affect clustering performance. To address this limitation, we are going to explore automatic parameter selection or parameter-free approaches in future work to enhance the robustness of the algorithm.

We hope that the revised manuscript addresses all comments and suggestions effectively. Thank you for your consideration.

Sincerely,

Qin Li and Geng Yang

The Shenzhen Institute of Information Technology

---

## [Decision Letter · Decision Letter 1]

26 Jan 2025

PONE-D-24-45409R1Multi-view Clustering via Global-view Graph LearningPLOS ONE

Dear Dr. Li,

Thank you for submitting your manuscript to PLOS ONE. After careful consideration, we feel that it has merit but does not fully meet PLOS ONE’s publication criteria as it currently stands. Therefore, we invite you to submit a revised version of the manuscript that addresses the points raised during the review process.

We look forward to receiving your revised manuscript.

Kind regards,

Wen Li

Academic Editor

PLOS ONE

Journal Requirements:

Reviewers' comments:

Reviewer's Responses to Questions

**Comments to the Author**

1. If the authors have adequately addressed your comments raised in a previous round of review and you feel that this manuscript is now acceptable for publication, you may indicate that here to bypass the “Comments to the Author” section, enter your conflict of interest statement in the “Confidential to Editor” section, and submit your "Accept" recommendation.

Reviewer #1: All comments have been addressed

Reviewer #3: (No Response)

2. Is the manuscript technically sound, and do the data support the conclusions?

Reviewer #1: Yes

Reviewer #3: Partly

3. Has the statistical analysis been performed appropriately and rigorously? 

Reviewer #1: Yes

Reviewer #3: Yes

4. Have the authors made all data underlying the findings in their manuscript fully available?

Reviewer #1: Yes

Reviewer #3: Yes

5. Is the manuscript presented in an intelligible fashion and written in standard English?

Reviewer #1: Yes

Reviewer #3: Yes

6. Review Comments to the Author

Reviewer #1: This paper proposes an efficient multiview spectral clustering method that constructs a Global-view Graph to integrate complementary information from different views. Unlike standard spectral clustering, their method directly assigns cluster labels without post-processing, such as K-means. After revisions, this paper meets the criteria for acceptance.

Reviewer #3: This work studies a graph-based method for multiview spectral clustering. While the structure is clear, I have the following concerns:

1. On page 6, in Equation (11), how is (11) derived from (9)? The Frobenius norm of \( ||X||_F = \sqrt{\sum_{i,j} X_{ij}^2} \), not merely the sum of all elements.

2. On page 6, in Equation (12), are the weights \( W_v \) and \( W \) guaranteed to be nonnegative?

3. In Equations (17) and (6), what is the difference between \( S_i^T \cdot 1_n = 1 \) and \( s_i 1_n = 1 \)?

4. On pages 7 and 8, does the proof establish Lemma 1? Additionally, how does it show the convergence of Algorithm 1?

7. PLOS authors have the option to publish the peer review history of their article (what does this mean?). If published, this will include your full peer review and any attached files.

Reviewer #1: No

Reviewer #3: No

---

## [Author Response · Author response to Decision Letter 2]

7 Mar 2025

[Manuscript ID]: PONE-D-24-45409

[Title]: Multi-view Clustering via Global-view Graph Learning

[Authors]: Dr. Qin Li and Geng Yang

[Date]: 2024.11.23

Dear Editor,

We would like to thank the academic editor and reviewers for their constructive feedback on our manuscript. Below, we provide detailed responses to all comments and explain how we have addressed them in the revised manuscript.

Comments from Academic Editor:

Comment 1: Thank you for submitting your manuscript to PLOS ONE. After careful consideration, we feel that it has merit but does not fully meet PLOS ONE’s publication criteria as it currently stands. Therefore, we invite you to submit a revised version of the manuscript that addresses the points raised during the review process.

Response: Thank you for your consideration of our manuscript and for providing us with the opportunity to revise and resubmit. We appreciate the constructive feedback from the reviewers, which has helped us improve the clarity and rigor of our work. In response to the reviewers' comments, we have carefully revised our manuscript to address all the concerns raised.

Comments from Reviewer #1:

Comment 1: This paper proposes an efficient multiview spectral clustering method that constructs a Global-view Graph to integrate complementary information from different views. Unlike standard spectral clustering, their method directly assigns cluster labels without post-processing, such as K-means. After revisions, this paper meets the criteria for acceptance.

Response: Thank you for your positive feedback and for recognizing the contributions of our work. We sincerely appreciate your time and effort in reviewing our manuscript. Your constructive comments have been invaluable in improving the clarity and rigor of our research.

Comments from Reviewer #3:

Comment 1: On page 6, in Equation (11), how is (11) derived from (9)? The Frobenius norm of \( ||X||_F = \sqrt{\sum_{i,j} X_{ij}^2} \), not merely the sum of all elements.

Response: Thank you for your valuable comment. There was indeed a writing error in the derivation, leading to an inaccurate expression of the Frobenius norm. The norm should be defined based on the sum of squared elements rather than merely summing all elements. I have carefully reviewed and corrected the derivation of Equation (11) from Equation (9) to ensure its accuracy. I sincerely appreciate your thorough review and insightful feedback, which have greatly helped improve the clarity and correctness of the paper. (Please refer to Section 'Optimization').

Comment 2: On page 6, in Equation (12), are the weights \( W_v \) and \( W \) guaranteed to be nonnegative?

Response: Thank you for your insightful comment. Equation (12) is related to Equation (11), but due to a writing error in Equation (11), the formulation of Equation (12) was also affected. As a result, the nonnegativity of the weights was not guaranteed. I have carefully reviewed and corrected both equations to ensure their correctness. I sincerely appreciate your thorough review and valuable feedback.(Please refer to Section ''Optimization'').

Comment 3: In Equations (17) and (6), what is the difference between \( S_i^T \cdot 1_n = 1 \) and \( s_i 1_n = 1 \)?

Response: Thank you for your valuable feedback regarding the constraints in Equations (6), (7), (8), and (17). I acknowledge that there was an inconsistency in the notation used to represent the row-wise normalization condition due to an oversight in the original formulation. Following your suggestion, I have carefully revised these four equations to ensure consistency in their expression. I sincerely appreciate your insightful comments, which have helped improve the clarity and precision of the paper. (Please refer to the revised section “Parameter-Weighted Multi-View Clustering (PwMC)” and “Self-Weighted Multi-View Clustering (SwMC)” in the manuscript for details.)

Comment 4: On pages 7 and 8, does the proof establish Lemma 1? Additionally, how does it show the convergence of Algorithm 1?

Response: Thank you for your insightful comment. Based on your guidance, we have carefully revisited and rederived the entire proof to ensure that Lemma 1 is properly established and to clarify how it demonstrates the convergence of Algorithm 1. We sincerely appreciate your valuable feedback, which has helped us improve the rigor and clarity of our work. (Please refer to the revised section “Converge Analysis” in the manuscript for details.)

Dear Editor and Reviewers, we sincerely appreciate your valuable feedback, which has helped us improve the manuscript. We hope that our revisions effectively address all comments and suggestions. Thank you for your time and consideration.

Sincerely,

Qin Li and Geng Yang

The Shenzhen Institute of Information Technology

---

## [Editor Report · Decision Letter 2]

10 Mar 2025

Multi-view Clustering via Global-view Graph Learning

PONE-D-24-45409R2

Dear Dr. Li,

We’re pleased to inform you that your manuscript has been judged scientifically suitable for publication and will be formally accepted for publication once it meets all outstanding technical requirements.

Kind regards,

Wen Li

Academic Editor

PLOS ONE

Additional Editor Comments (optional):

This presentation has made modifications based on the referees' report. So I recommend accepting the revised version for publication on PLOS One
---

## [Editor Report · Acceptance letter]

PONE-D-24-45409R2

PLOS ONE

Dear Dr. Li,

I'm pleased to inform you that your manuscript has been deemed suitable for publication in PLOS ONE. Congratulations! Your manuscript is now being handed over to our production team.

Kind regards,

on behalf of

Dr. Wen Li

Academic Editor

PLOS ONE